# Triptolide Reduces Neoplastic Progression in Hepatocellular Carcinoma by Downregulating the Lipid Lipase Signaling Pathway

**DOI:** 10.3390/cancers16030550

**Published:** 2024-01-27

**Authors:** Wei Chang, Jingjing Wang, Yuanqi You, Hongqian Wang, Shendong Xu, Stephen Vulcano, Changlu Xu, Chenlin Shen, Zhi Li, Jie Wang

**Affiliations:** 1Inflammation and Immune Mediated Diseases Laboratory of Anhui Province, School of Pharmacy, Anhui Medical University, Hefei 230032, China; changw@bbmc.edu.cn (W.C.); cyyuanqi@163.com (Y.Y.); xushendongdev@126.com (S.X.); 2Anhui Engineering Technology Research Center of Biochemical Pharmaceuticals, Faculty of Pharmacy, Bengbu Medical College, Bengbu 233030, China; 3Department of Pathology and Gastroenterology, The First Affiliated Hospital of Anhui Medical University, Hefei 230032, China; wangjjhq@163.com (J.W.); ahykdxwanghongqian@163.com (H.W.); 4Autoimmunity and Inflammation Program, HSS Research Institute, Hospital for Special Surgery, New York, NY 10021, USA; vulcanos@hss.edu; 5Division of Oral and Systemic Health Sciences, School of Dentistry, The University of California, Los Angeles, CA 90095, USA; changluxu@ucla.edu (C.X.); lizhi7@ucla.edu (Z.L.)

**Keywords:** hepatocellular carcinoma, triptolide, lipid lipase, lipid metabolism

## Abstract

**Simple Summary:**

TP is a widely utilized anticancer medication, particularly effective in treating HCC. Thorough investigation into TP’s molecular mechanism in HCC treatment is crucial for precise and combination therapy. In this paper, we have unveiled a novel mechanism of TP in HCC treatment, where it inhibits tumor growth by reducing lipid accumulation. While the inhibition of P53 gene activity is commonly thought to be the reason for TP’s effectiveness in treating HCC, this new mechanism serves as a significant complement to the current understanding, paving the way for innovative approaches to HCC treatment.

**Abstract:**

Hepatocellular carcinoma (HCC), which is the third leading cause of cancer-related mortality in the world, presents a significant medical challenge. Triptolide (TP) has been identified as an effective therapeutic drug for HCC. However, its precise therapeutic mechanism is still unknown. Understanding the mechanism of action of TP against HCC is crucial for its implementation in the field of HCC treatment. We hypothesize that the anti-HCC actions of TP might be related to its modulation of HCC lipid metabolism given the crucial role that lipid metabolism plays in promoting the progression of HCC. In this work, we first demonstrate that, both in vitro and in vivo, TP significantly reduces lipid accumulation in HCC cells. Additionally, we notice that lipoprotein lipase (LPL) expression is markedly upregulated in HCC, and that its levels are positively connected with the disease’s progression. It is interesting to note that TP dramatically reduces LPL activity, which in turn prevents HCC growth and reduces lipid accumulation. Additionally, the effect of TP on LPL is a direct correlation. These results definitely demonstrate that TP protects hepatocytes against abnormal accumulation of lipids by transcriptionally suppressing LPL, which reduces the development of HCC. This newly identified pathway provides insight into the process through which TP exerts its anti-HCC actions.

## 1. Introduction

Hepatocellular carcinoma (HCC), as the most prevalent primary liver malignancy, is the third leading cause of cancer death worldwide [1]. A significant burden is placed on both those who are affected and healthcare systems as a whole by their low relative 5-year survival rate of only 18% [2]. Fortunately, because of hepatitis vaccinations, the frequency of malignant tumors connected to viral hepatitis has gradually decreased in recent years. [3,4]. However, non-alcoholic steatohepatitis (NASH) and non-alcoholic fatty liver disease (NAFLD)-related liver cancer cases are on the rise [5]. NAFLD is estimated to affect one in four people worldwide, and, given the present obesity epidemic, it is anticipated that the prevalence of NAFLD-related HCC would increase [6].

The molecular pathways underlying the progression from NAFLD to NASH and then HCC have been well studied during the last few decades [6,7]. Research in this field has demonstrated the significance of lipid metabolism issues in the emergence of NAFLD-induced HCC [8,9]. NAFLD can ultimately result in cirrhosis and HCC, with an intermediate step defined by reversible intracellular accumulation of lipids developing into inflammation and/or fibrosis [10]. Consistently, epidemiological data indicate that obesity increases the risk of HCC [11]. Similarly, it has been demonstrated that factors that promote accumulation of fat, such as diabetes and obesity, make people more susceptible to HCC [12,13]. For tumor cell proliferation, fatty acids (FA), which include both exogenous uptake and de novo synthesis, serve as the primary source of energy [14]. Fatty acid synthetase (FASN), which produces long-chain fatty acids from acetyl-CoA and malonyl-CoA, serves as crucial for endogenous fatty acid synthesis [15]. As a result, blocking fatty acid synthase has been proposed to be a promising and effective strategy to prevent tumor growth, as past studies have shown that FASN inhibitors are effective in this regard [16]. Notably, circulating lipoprotein particles containing triglycerides can act as a major source of energy for tumor cell growth when exogenous fatty acids are employed as a supplement [8]. Lipoproteins are classified into five categories based on their density and size, namely very low-density lipoproteins (VLDLs), low-density lipoproteins (LDLs), intermediate-density lipoproteins (IDLs), high-density lipoproteins (HDLs), and chylomicrons (CM) [17]. These different types of lipoproteins exhibit varying ability in the binding of triglycerides and cholesterol. HDL, LDL, and CM primarily facilitate the transportation of cholesterol, while IDL and VLDL predominantly transport triglycerides [18]. Circulating triglycerides must first be hydrolyzed by extracellular lipoprotein lipase (LPL) before being used by tumor cells [19]. Therefore, preventing LPL activity may reduce proliferation of tumor cells by limiting the energy source. Unfortunately, there has not been much study performed regarding the use of LPL inhibitors in HCC. As a result, it is urgently necessary to understand the probable processes behind the advancement of lipid metabolism in HCC and investigate effective therapy options.

Triptolide (TP), a diterpene active molecule extracted from medicinal plants and then purified from other chemicals, was first found in 1972 by Kupchan et al. It was discovered that TP has an exceptional anti-leukemia effect [20]. In addition, other important physiological effects of TP, such as anti-inflammatory, antioxidant, immunosuppressive, anticancer, have been scientifically demonstrated [21,22,23,24]. According to research, combining Sorafenib with TP can improve the therapeutic efficacy for HCC [25]. Additionally, through the NF-B signaling pathway, TP prevents the invasion and carcinogenesis of HCC/MHCC-97H cells [26]. Despite the fact that the underlying processes are still largely unknown, these findings indicate to the therapeutic potential of TP in the treatment of HCC. In previous investigation, we found that the tripterygium wilfordii extract Celastrol inhibits the growth of HCC caused by Diethylnitrosamine (DEN) [27]. However, the function of TP, another tripterygium wilfordii extract, in DEN or carbon tetrachloride (CCl_4_)-induced liver cancer has not been clarified. Furthermore, it is still unclear whether TP may prevent tumor cell proliferation by interfering with lipid metabolism.

In the present study, we found that the increase in LPL expression of HCC was associated with the prognosis of disease. DEN and CCl_4_-induced HCC, as well as the growth of xenograft tumors, were significantly suppressed by TP in a dose-dependent manner. This inhibitory impact may be related to LPL inhibition, which prevents tumor cells from uptaking exogenous fatty acids. Furthermore, tripterygium’s growth-inhibitory effect on xenograft tumors was amplified by decreased LPL expression, which further inhibited xenograft tumor growth. According to research, TP selectively binds with p53 and encourages apoptosis in hepatoma cells. These findings would demonstrate that TP has the potential to be an HCC drug and provide insight into the mechanisms through which TP prevents HCC.

## 2. Materials and Methods

### 2.1. Human HCC Samples and Cell Lines

HCC and para-tumor tissue samples were surgically resected from HCC patients who underwent hepatectomy at the first affiliated hospital of Anhui Medical University (Anhui, China) between 2021 and 2022. The normal serums were collected from health examinees at the same hospital. All diagnoses were confirmed by pathology. Complete clinicopathological and follow-up data were available for the 30 HCC samples. Then, 30 HCC samples and para-tumor tissue samples were used for quantitative real-time PCR (qRT-PCR) analysis. Clinical characteristics of the study cohort are provided in Appendix A. This study was approved by the Ethics Committee of Anhui Medical University. HCC cell lines SMMC7721 and HepG-2 (Institute of Biochemistry and Cell Biology of the Chinese Academy of Sciences, Shanghai, China) were used in the following studies. HepG2 is a highly differentiated liver cancer cell line used as the experimental group, while SMMC-7721 is a poorly differentiated liver cancer cell line used as the control group. Both cell models were utilized to jointly validate the therapeutic effect of TP on liver cancer cells.

Cells were cultured in RPMI-1640 or DMEM medium supplemented with 10% fetal bovine serum (FBS, Hyclone, Logan, UT, USA) and 1% penicillin/streptomycin (P/S) in a humidified atmosphere with 5% CO_2_ at 37 °C. 

### 2.2. Natural Small Molecule

Triptolide (C_20_H_24_O_6_, molecular weight 360.4) was obtained from MedChemExpress (Monmouth Junction, NJ, USA). The purity was 99.86%, which was identified by high-performance liquid chromatography.

### 2.3. Animals

Male C57BL/6J mice (6 weeks of age) were purchased from the Laboratory Animal Centre of Anhui Medical University. These mice were fed in a standard vivarium with 12 h light/12 h dark cycles and had free access to food and water. This study was approved by the Ethics Review Committee for Animal Experimentation of the Institute of Clinical Pharmacology, Anhui Medical University, China. After 1 day of acclimating, mice were randomly divided into five groups, which include the control group (normal), DEN-treated (model), L-TP (5 mg/kg), M-TP (10 mg/kg), and H-TP (20 mg/kg) groups. DEN-treated mice were injected intraperitoneally (i.p.) with DEN (2 mg/kg, Sigma-Aldrich, St. Louis, MO, USA) at 15 days, and then were administered intraperitoneally with CCl_4_ (5 mL/kg, 20%) once a week for 15 weeks. On the ninth week, the mice in TP groups were treated intragastrically with different doses of TP until the end of the experiment (19 weeks). On the day after the last administration, the body weights of the mice were measured, and the blood sample was taken by removing an eyeball, and the liver specimens were harvested and then weighed. 

### 2.4. Cell Proliferation and Colony Formation Assay

The Cell Counting Kit-8 (CCK-8) was used to measure cell viability according to the manufacturer’s instructions, and the quantification was conducted by using a Spectra Rainbow microtiter plate reader (Tecan, Kawasaki, Japan) and the Genetix Clone Select Imager System. For colony formation assay, cells were plated in six-well plates with 1 × 10^3^ cells per well, and the medium was changed every 3 days. After 2 weeks, colonies were fixed with 4% paraformaldehyde and stained with 0.1% crystal violet for 20 min. Each experiment was repeated at least three times.

### 2.5. Transwell Migration Assay

Transwell chambers (24-well plate, 8 μm pores, BD Biosciences, Dubai, United Arab Emirates) were used for double-chamber migration assays. Briefly, 600 μL of DMEM was injected into the lower chambers. HCC cells with different treatments were suspended in the serum-free medium, were seeded in the upper chambers, and incubated for 24 h. Then, the cells on the upper surface of the filters were removed using cotton wool swabs. The migrated cells on the lower side were fixed in 95% methanol and stained with 0.1% crystal violet dye, and the number of cells which migrated on the lower surface were counted in three randomly selected high-magnification fields (100×) for each sample.

### 2.6. Cell Transfection

Plasmids expressing HA-tagged LPL (pcLPL) were constructed by inserting full-length human LPL cDNA into the pcDNA3.0 (Invitrogen, Waltham, MA, USA). pGL3-LPLp luciferase reporter plasmid was constructed by inserting the promoter of the LPL gene into pGL3-Basic vector (Promega, Tokyo, Japan). All recombinant plasmids were confirmed by DNA sequencing. Cells were transfected with the indicated plasmids using Lipofectamine™ 2000 (Thermo Fisher Scientific, Waltham, MA, USA) according to the manufacturer’s protocol.

### 2.7. Quantitative Real-Time PCR (q-RT-PCR)

Total RNA from liver tissues were extracted by using TRIzol (Invitrogen) and then transcribed in cDNA by the PrimeScript™ 1st Strand cDNA Synthesis Kit (TaKaRa, Kusatsu, Japan). Subsequently, the q-RT-PCR reactions were performed by using the Light-Cycler system (Roche Applied Science, Penzberg, Germany) and the expression of selected genes was analyzed by using the LightCycler 480 System (Roche, Basel, Switzerland) and SYBR Green chemistry. All PCR data were presented relative to the mean of housekeeping genes (2^−ΔΔCt^ method). Gapdh was used for normalization mRNA levels. Primers are listed, mLpl (Fwd: 5′-ATGGATGGACGGTAACGGGAA-3′; Rev 5′-CCCGATACAACCAGTCTACTACA-3′); Gapdh (Fwd: 5′-ACTCCACTCACGGCAAATTC-3′; Rev 5′-CCTTCCACAATGCCAAAGTT-3′).

### 2.8. Western Blot Assay (WB)

Cell lines or tumor tissues were collected in RIPA buffer with protease inhibitor cocktails (Thermo Fisher Scientific) and lysed on ice for 30 min. Lysates were centrifuged for 15 min at 13,000× *g* and 4 °C, and the supernatants were collected; then, their protein concentrations were determined by the BCA Protein Assay Reagent (Thermo Fisher Scientific). Protein samples were transferred onto a PVDF membrane which was incubated with a 1:1000 dilution of primary antibodies overnight at 4 °C. Next, the membrane was incubated with HRP-conjugated rabbit or mouse secondary antibodies at room temperature for 1.5 h and developed using a chemiluminescence reagent (Advansta, San Jose, CA, USA). The commercial antibodies used in this study included anti-LPL (1:1000; Proteintech, Rosemont, IL, USA), anti-β-actin (1:5000; Cell Signaling Technology, Danvers, MA, USA), anti-p53 (1:1000; Abcam, Cambridge, UK), anti-MDM2 (1:1000; Abcam), anti-Bax (1:1000; Abcam), anti-Bcl-2 (1:1000; Abcam), and anti-Bcl-xl (1:1000; Abcam).

### 2.9. Immunohistochemistry (IHC)

The sections were deparaffinized and rehydrated through xylene/absolute alcohol. Endogenous peroxidase activity was blocked by incubating the sections in methanol with 0.6% hydrogen peroxide. Subsequently, 3% H_2_O_2_ and goat serum were used to treat the samples; then, the samples were incubated with anti-LPL overnight at 4 °C, followed by incubation with a secondary antibody, and visualized using DAB substrate. IPP6.0 software was utilized to carry out quantification of immunohistochemistry staining.

### 2.10. In Vivo Tumor Growth Assay

Sh-ctrl-, sh-LPL-, and TP-treated cells were injected subcutaneously into the flanks of nude mice (eight per group). A total of 1 × 10^6^ cells were injected subcutaneously into female BALB/c nude mice (Hangzhou Ziyuan Laboratory Animal Science and Technology). Tumor volumes were measured every 4 days and calculated by the formula (length × width^2^)/2. After four weeks, the tumor tissues were removed and weighed.

### 2.11. Statistical Analysis

For statistical analysis, SPSS 22.0 and GraphPad Prism 7.0 software were used. Values are shown as the mean ± the standard error of the mean (S.E.M). Analyses between groups were performed using one-way ANOVA or the two-tailed Student’s *t*-test. A *p* value < 0.05 was considered a statistically significant result.

## 3. Results

### 3.1. High LPL Expression Predicted an Unfavorable Prognosis in HCC Patients

It has been confirmed that lipid metabolic dysfunction plays a significant role in the development of HCC. Here, we showed the overall area of LDs tends to increase in HCC patients’ malignant tissues (Figure 1a,b). LPL is a key factor in the development of metabolic abnormalities. LPL alone can induce LDL aggregation, similar to the mechanism of lipoprotein lipase and sphingomyelin enzyme in low-density lipoprotein aggregation. LPL hydrolyzes triglycerides to produce small dense particles, which then transform into LDL in the peripheral circulation. Our research revealed that serum HDL levels were significantly lower in HCC patients, while serum VLDL and LDL levels were significantly elevated (Figure 1d). Analysis of the TCGA database indicated that the expression of LPL mRNA was significantly increased in malignant tissues from HCC patients compared with para-cancerous tissues (Figure 1f), and a higher LPL expression was significantly associated with poor patient prognosis (Figure 1c). Furthermore, we observed a strong association between HCC and LPL expression (Figure 1e). On the contrary, we also conducted a statistical analysis of the levels of ANGPTL3, PNPLA2, LIPE, and LIPC in the patients’ blood (Appendix A), and the results showed that there was no correlation between these lipid components and HCC. These findings suggest that the development of HCC may be linked to abnormal lipoprotein metabolism driven by increased LPL expression in liver tissues.

### 3.2. Triptolide Ameliorates DEN- and CCl_4_-Induced HCC in Mice

For the purpose of evaluating triptolide’s pharmacological effects on liver damage caused by carbon tetrachloride (CCl_4_) and diethylnitrosamine (DEN), we first set out to analyze in vivo liver damage caused by these two chemicals. The animal experiment was designed and carried out as shown in Figure 2a, and Appendix A shows the mass spectrometry analysis of TP to elucidate its structure. Hepatocarcinogenesis demonstrated that triptolide significantly reduced DEN- and CCl_4_-induced changes to the liver’s morphology (Figure 2b). From a morphological perspective, compared with the control group, we can clearly observe that the size and number of tumor nodules in the TP treatment group decreased significantly, and this change reappeared in a dose-dependent manner. From a histological perspective, we also found that the degree and area of vacuolization in the tumor tissue decreased significantly with increasing TP dose. The three primary diagnostic indicators utilized in liver function tests, AST, ALT, and AFP, were significantly reduced in the triptolide group than in the model group, indicating that the liver functions had returned to normal (Figure 2c–e). Additionally, BODIPY staining showed that triptolide greatly decreased the amount of lipid droplets that were deposited (Figure 2f,g). These findings show that triptolide reduced the liver damage and HCC caused by DEN and CCl_4_.

### 3.3. Triptolide Triggers HCC Cells Apoptosis by LPL and p53-Bax Pathways in Mice

We used immunohistochemistry and Western blot to verify the effect of TP on LPL. Figure 3a shows the results of immunohistochemistry, with significant yellow areas in the model group indicating high expression of LPL. Compared with this, the TP-treated group showed a significant decrease in LPL expression and exhibited a significant dose-dependent effect. Among them, the level of LPL in the high concentration TP treatment group was close to the level of normal liver. The Western blot results further confirmed the above results (Figure 3b). Proteins positively correlated with LPL activity, such as MDM2, Bcl-2, and Bcl-xl, were significantly inhibited, while proteins that inhibited HCC activity, including p53 and Bax, showed significant improvement. In vivo, triptolide increased the expression of BAX, a pro-apoptotic factor, while downregulating the expression of Bcl-2 and Bcl-xl, two anti-apoptotic protein indicators (Figure 3c–f). These findings show that in mice, Triptolide induces apoptosis of HCC cells via the LPL and p53-Bax pathways

### 3.4. Triptolide Inhibits HCC Cell Viability and Proliferation, and Promotes HCC Cell Apoptosis by Suppressing Expression of LPL

Triptolide was applied to the SMMC7721 and HepG2 cells at various concentrations for 24 and 48 h, after which its anticancer effects were evaluated. Triptolide administration resulted in a dose- and time-dependent decrease in cell viability in both SMMC7721 and HepG2 cells, as well as an increase in apoptosis (Figure 4a–d). Furthermore, Western blotting analysis revealed that triptolide therapy reduced the protein expression of LPL (Figure 4e–h). These findings suggested that triptolide increases LPL protein degradation in order to initiate apoptosis in HCC cells.

### 3.5. Inhibition of LPL Decreased HCC Cell Proliferation and Invasion

Cell line HepG2 was selected for further investigation because they showed the highest expression of LPL, demonstrating that LPL is the predominant target protein for the apoptosis produced by triptolide in HCC cells. HepG2 cells with stable LPL knockdown were established (sh-LPL cells, Figure 5a). When compared with sh-ctrl cells, LPL inhibition significantly reduced cell growth (Figure 5b). Sh-LPL significantly suppressed colony formation, according to a colony formation experiment (Figure 5c,d). LPL knockdown suppresses HepG2 invasion (Figure 5e,f).

### 3.6. Triptolide Suppressed Xenograft Tumor Growth of HCC Cells In Vivo via Targeting LPL

Nude mice bearing established HepG2-sh-ctrl or HepG2-sh-LPL xenografts were utilized to test if LPL is the primary target protein of triptolide in vivo. LPL knockdown was employed to test the anti-tumor activity of triptolide. When tumors reached a size of less than 50 mm^3^, mice were given triptolide at a dose of 2 mg/kg/d five days a week, with the NS (vehicle)-treated mice serving as the control group. Triptolide decreased HepG2-sh-ctrl xenograft tumor growth overall, decreasing tumor size and tumor volume. LPL knockdown group also had a comparable impact (Figure 6a,b). Furthermore, triptolide treatment’s effects on tumor growth inhibition decreased when LPL was knocked down (Figure 6c,d). These findings show that triptolide targets LPL to inhibit the formation of HCC cells’ xenograft tumors in vivo.

## 4. Discussion

Disorders of lipid metabolism can result in liver diseases including non-alcoholic fatty liver disease and hepatocellular carcinoma because the liver is the primary site of lipid metabolism [9,28]. Hepatocellular carcinoma cells have been shown to utilize both endogenous and exogenous FA to support their own proliferation [8]. It is interesting to note that the uptake of exogenous FA strongly depends on triglyceride LPL breakdown [8]. In fact, LPL is overexpressed in hepatocellular carcinoma, non-small cell lung cancer (NSCLC), and chronic lymphocytic leukemia (CLL) [19,29,30]. Further evidence that LPL is a poor prognostic indicator for human hepatocellular carcinoma comes from the association of high levels of LPL with aggressive tumor morphologies and patient short-term survival. Consequently, a potential therapeutic for inhibiting the proliferation of liver cancer cells involves targeting LPL. We observed in this study that LPL expression was dramatically increased in HCC and associated with the prognosis of the condition. It was discovered that the growth of DEN- and CCl_4_-induced HCC, as well as transplanted tumors, was dose dependently inhibited by Marsdenia tenacissima. In addition, the transplanted tumor’s growth was slowed down by reduced expression of LPL, which also strengthened tripterygium’s ability to stop the tumor’s growth. These results suggest that TP may be an effective treatment for hepatocellular carcinoma and that LPL is a potential target for the treatment of the disease.

TP was isolated from tripterygium wilfordii, a perennial Chinese herb [20]. TP has received extensive research because of its significant contribution to the pharmacological actions that include antibacterial, anti-inflammatory, anti-rheumatic, immune regulatory, and anticancer [24,31,32]. Due to its capacity to eradicate cancer cells from a variety of organs, including the colon, ovaries, breast, blood, brain, prostate, and kidney, TP has a broad spectrum of antitumor action [33]. Multiple studies have demonstrated that TP suppressed HCC with the support of various vectors [34,35]. However, not much study has been performed on the mechanism through which TP inhibits HCC. In the current work, mice treated with DEN and CCl_4_ developed HCC, demonstrating the viability of the HCC model. Importantly, tumor volume decrease showed that TP therapy suppressed HCC in a dose-dependent manner over the final six weeks of the trial. The serum transaminases, such as aspartate transaminase (AST) and alanine transaminase (ALT), are routine laboratory markers for liver function [36]. Additionally, after receiving TP therapy, the liver function indicators ALT and AST considerably improved. As a serum indicator of primary liver cancer, AFP can be used to diagnose and track the effectiveness of HCC treatments [37]. AFP considerably increased in the model group, but TP dose dependently decreased serum AFP content, demonstrating TP’s anti-HCC activity. Furthermore, in a xenograft tumor model caused by HepG2, TP dramatically reduced tumor growth. The CCK8 test was used to assess the killing impact of various concentrations of TP on the human HCC cell lines SMMC7721 and HepG2 in order to ascertain the effect of TP in liver cancer cells. We found that TP decreased both cells’ ability to survive in a dose-dependent way. Experiments with colony formation revealed that TP greatly reduced the ability of HepG2 cells to form colonies. Transwell migration assay results showed that TP treatment greatly reduced HepG2 cells’ capacity for invasion. These findings supported TP’s ability to prevent HCC growth. 

LPL, a lipid metabolism rate-limiting enzyme, is primarily produced in adipose tissue, skeletal muscle, and the mammary gland. It can facilitate the breakdown of circulating lipoproteins high in triglycerides, such as CM, LDL, and VLDL [38]. Diseases connected to lipid metabolism, like hyperlipidemia and fatty liver disease, are impacted significantly by LPL. LPL has also been demonstrated to support the development and survival of breast cancer cells [39]. The development of non-alcoholic steatohepatitis into liver cancer was substantially prevented by targeted suppression of LPL-mediated fatty acid metabolism [40]. In any event, relatively little study has been conducted regarding the involvement of LPL in HCC. In the TCGA database, we first looked for a correlation between LPL and the prognosis for HCC. It is noteworthy that LPL was significantly expressed in HCC patients and was linked to a poor prognosis for the condition. The expression level of LPL was also much higher in HCC tumor tissues than in paracancer tissues, as shown by IF and IHC. Intriguingly, serum levels of TG, LDL, and VLDL were considerably greater in HCC patients than in healthy controls, pointing to a potential rise in exogenous fatty acid consumption. Previous research has demonstrated that blocking LPL prevented a variety of hepatoma cells from proliferating [8,30]. We observed that sh-LPL greatly reduced the capacity of HepG2 cells to proliferate and invade, which is consistent with other research. Additionally, we discovered that sh-LPL prevented HCC xenograft tumor growth. The aforementioned findings supported LPL’s inhibitory effect on HCC, although it is still unclear how LPL contributes to TP’s ability to suppress HCC. It is interesting to note that both in vivo and in vitro TP preconditioning suppressed LPL expression in a dose-dependent manner. We also discovered that sh-LPL improved the inhibitory effect of TP on HCC cells’ capacity to proliferate and invade. Similar to this, in HCC xenograft tumor models, sh-LPL also improved the inhibitory impact of TP on the tumor. This suggested that TP might prevent tumor cells from taking exogenous fatty acids by inhibiting LPL.

Over 60% of human malignancies include P53 deletions or missense mutations, which play a significant role in the genesis of cancer [41]. P53 activation reduced tumor cell growth by increasing the expression of apoptotic proteins Bax [42,43]. In our research, TP dose dependently suppressed the production of the anti-apoptotic proteins Bcl-2 and Bc-xl while promoting the expression of p53 and Bax. These findings revealed that TP caused HCC cells to undergo apoptosis by activating p53.

However, several limitations remain in the present study. First, studies have demonstrated that high doses of TP can produce serious side effects, such as eleukopenia, urinary abnormalities, liver damage, and myocardial damage [33]. However, we did not consider the systemic toxicity of TP in the present study. Research has found that synergetic delivery of TP and Ce6 with light-activatable liposomes have higher safety in the treatment of HCC [44]. Therefore, it is necessary to develop new delivery methods for the treatment of HCC in the future study. Second, TP is often used as an adjunct to HCC immunotherapy, such as in combination with sorafenib. An earlier study found that the combination of sorafenib and triptolide had shown synergistic effects on HCC [25]. However, due to the difference in pharmacokinetic process and biological distribution between sorafenib and TP, the proportion of drug distribution in tumor sites is uncontrollable, which especially limits the clinical application of TP. To overcome this problem, Li et al. constructed a biomimetic nanosystem based on cancer cell–platelet hybrid membrane camouflage to co-deliver sorafenib and TP [45]. However, this study did not observe the effect of TP combined with sorafenib on the therapeutic efficacy of HCC. In addition, the mode of administration still needs to be further optimized in follow-up studies.

## 5. Conclusions

In conclusion, TP inhibits the production of LPL by specific interaction with p53, hence preventing the uptake of foreign lipids by HCC cells and preventing tumor formation. Additionally, TP causes the death of hepatoma cells, which has tumor-suppressive properties. Our research offers fresh understandings into the mechanisms underlying the development of HCC as well as an innovative strategy to treating HCC.

## Figures and Tables

**Figure 1 cancers-16-00550-f001:**
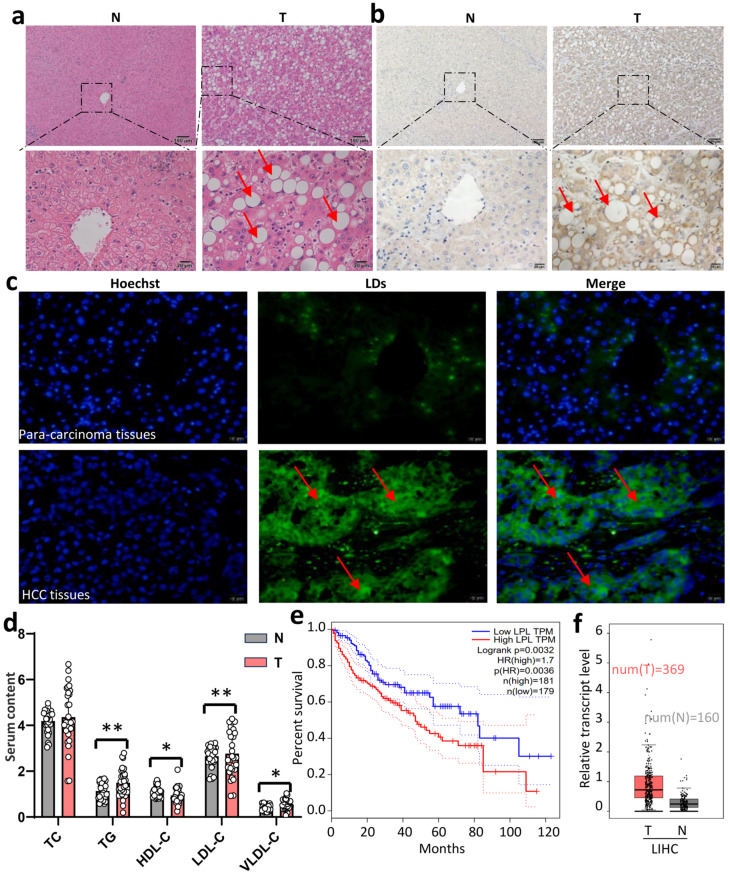
High expression of LPL in patients with HCC is associated with poor prognosis of the disease. (**a**) Representative pictures of HE staining of HCC and para-carcinoma tissue (10× and 40×). (**b**) Representative pictures of BODIPY staining in HCC and para-carcinoma tissue (10× and 40×). (**c**) IHC representative pictures of LPL in HCC and para-carcinoma tissue. (**d**) Statistical analysis of TC, TG, and lipoprotein in HCC and healthy controls (*n* = 30 in each group), the picture is enlarged by 40 times. (**e**) Overall survivals of HCC patients with high LPL and low LPL were analyzed using Logrank analysis. (**f**) LPL mRNA expression levels in HCC and para-carcinoma tissue in TCGA database. * *p* < 0.05, ** *p* < 0.01. N represents para-carcinoma tissues, and T represents HCC tissues. The inserted arrows indicate some distinctive changes.

**Figure 2 cancers-16-00550-f002:**
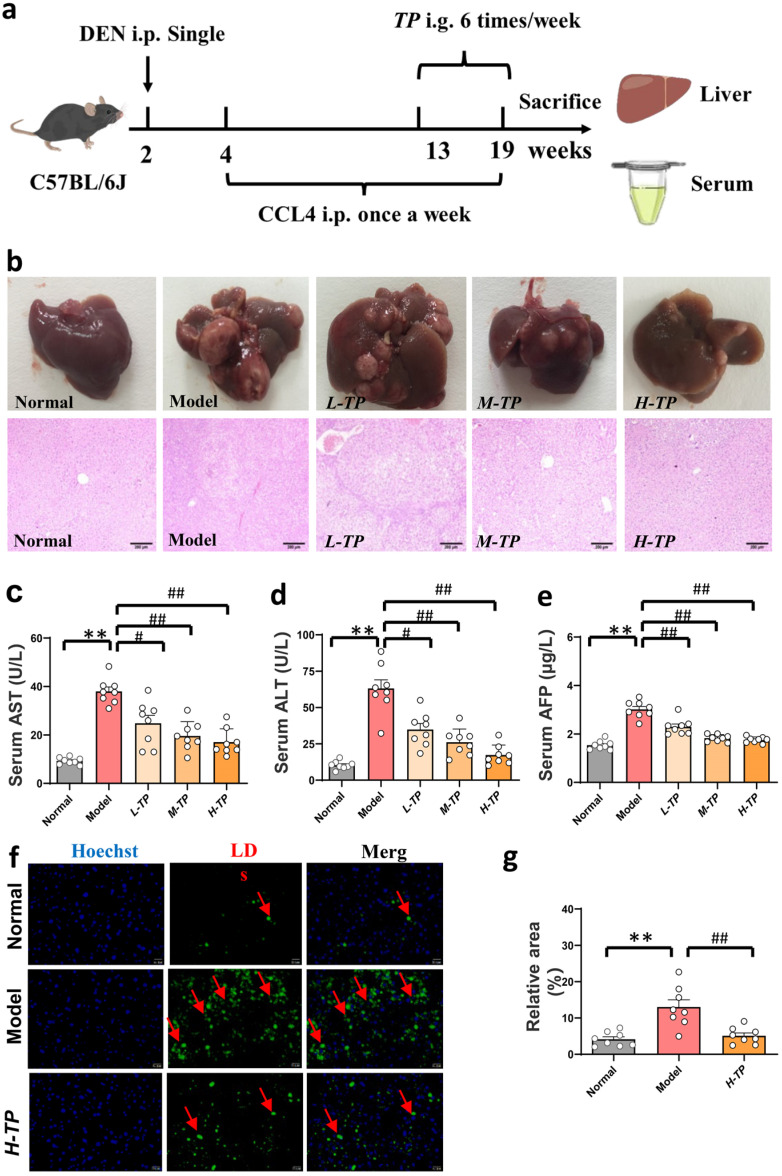
Triptolide ameliorates DEN- and CCl_4_-induced HCC in mice. After administering an intraperitoneal injection of DEN (2 mg/kg) at the second week, carbon tetrachloride (5 mL/kg, 20%) was injected intraperitoneally from 4 w to 19 w once a week to induce the HCC model. And then, from the ninth week, the mice in TP groups [L-TP (5 mg/kg)-, M-TP (10 mg/kg)-, and H-TP (20 mg/kg)] were treated intragastrically with different doses of TP until the end of the experiment (19 weeks). (**a**) Schematic diagram of establishment and intervention of mouse HCC model. (**b**) Effects of different doses of TP on DEN- and CCl_4_-induced HCC, and the representative images for the number of mouse liver nodules (upper) and H&E staining of orthotopic HCC tumor tissues (down), the picture is enlarged by 40 times. (**c**–**e**) Serum ALT, AST and AFP were measured in different groups (*n* = 8 in each group). (**f**) Representative pictures of BODIPY staining, the picture is enlarged by 400 times. The inserted arrows indicate some distinctive changes. (**g**) Quantitative analysis of BODIPY staining. (*n* = 8 in each group), ^#^
*p* < 0.05, **^/##^ *p* < 0.01.

**Figure 3 cancers-16-00550-f003:**
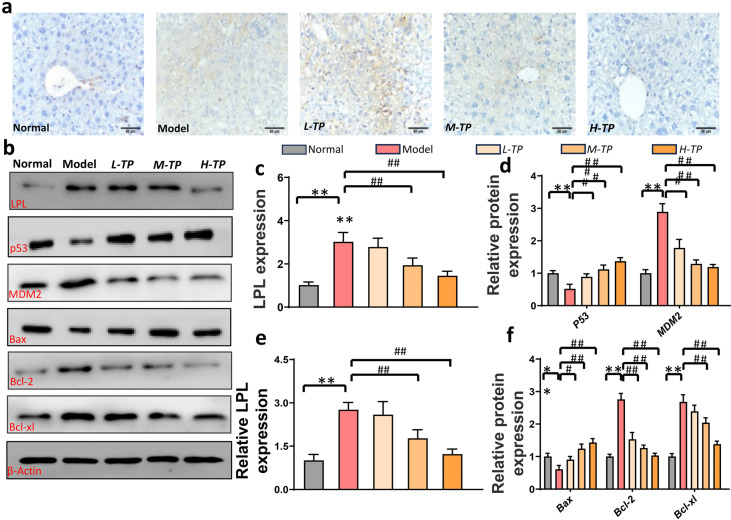
Triptolide triggers HCC cells apoptosis in mice. (**a**) IHC representative pictures of LPL in mice liver tissues(200×). (**b**) Hepatic LPL, p-53, MDM2, Bax, Bcl-2, and Bcl-xl were measured using Western blots. (**c**) Quantitative analysis of LPL in IHC (*n* = 8 in each group). (**d**) Quantitative analysis of p-53 and MDM2 from western blots (*n* = 6 in each group). (**e**) Quantitative analysis of LPL from western blots (*n* = 6 in each group). (**f**) Quantitative analysis of Bax, Bcl-2, and Bcl-xl from Western blots (*n* = 6). *^/#^
*p* < 0.05, **^/##^ *p* < 0.01.

**Figure 4 cancers-16-00550-f004:**
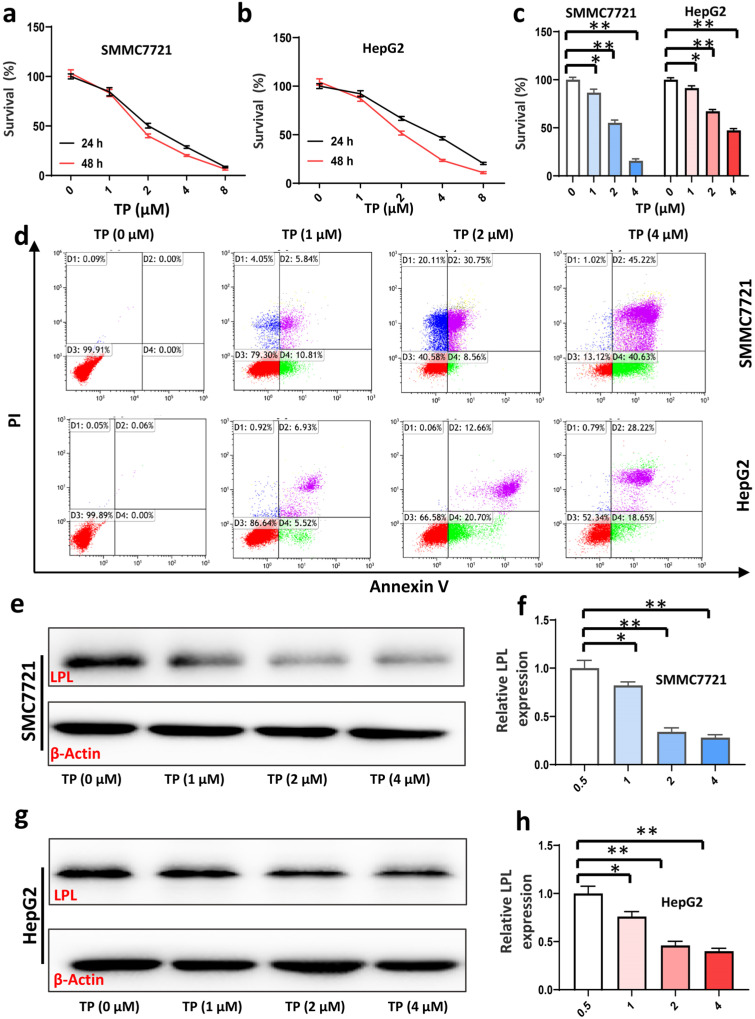
Triptolide inhibits the survival and proliferation of HCC cells and promotes apoptosis of HCC cells in vitro. (**a**,**b**) Effects of different doses of TP on proliferation of SMMC7721 and HepG2 cells. (**c**) Quantitative analysis of cell proliferation after triptolide treatment (*n* = 5 in each group). (**d**) The apoptosis of SMMC7721 and HepG2 cells were measured by flow cytometry. (**e**) LPL expressions of SMMC7721 after triptolide treatment were measured using Western blots. (**f**) Quantitative analysis of LPL of SMMC7721 (*n* = 6 in each group). (**g**) LPL expressions of HepG2 after triptolide treatment were measured using Western blots. (**h**) Quantitative analysis of LPL of HepG2 (*n* = 6 in each group). * *p* < 0.05, ** *p* < 0.01.

**Figure 5 cancers-16-00550-f005:**
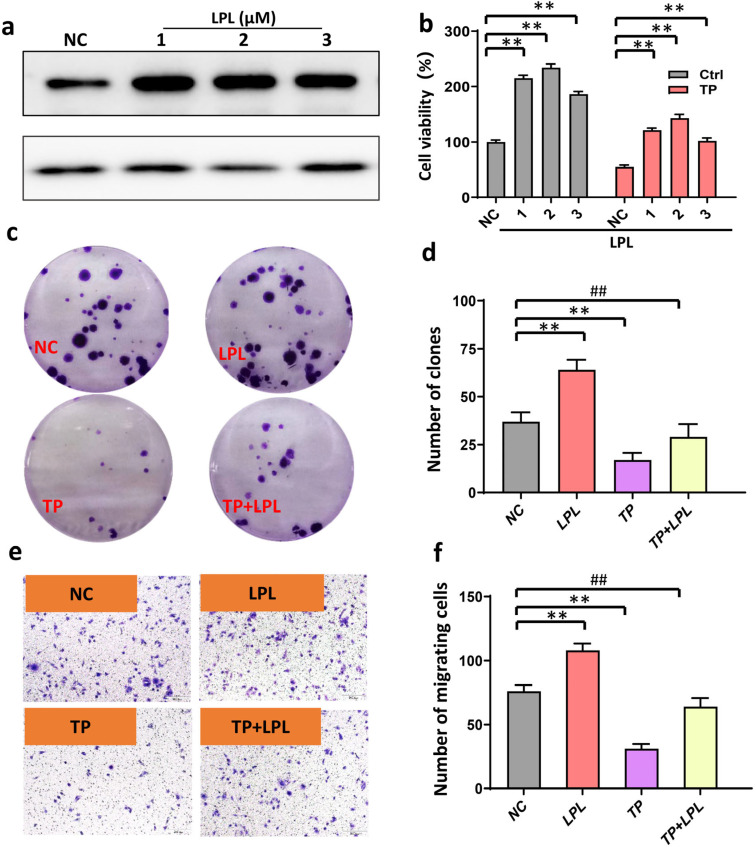
Overexpression of LPL promotes colony formation and invasion. (**a**) Representative bands were measured using Western blots after LPL overexpression. (**b**) Statistical analysis of overexpression of LPL on cell viability (*n* = 4 in each group). (**c**) Representative picture: effect of LPL overexpression on colony formation after TP treatment (no amplification). (**d**) Statistical analysis of the number of colonies in different groups (*n* = 6). (**e**) Representative picture: the effect of LPL overexpression on the inhibition of tumor invasion by TP (*n* = 8, 400×). (**f**) Statistical analysis of the number of migrating cells in different groups. ** *p* < 0.01. ^##^ *p* < 0.01.

**Figure 6 cancers-16-00550-f006:**
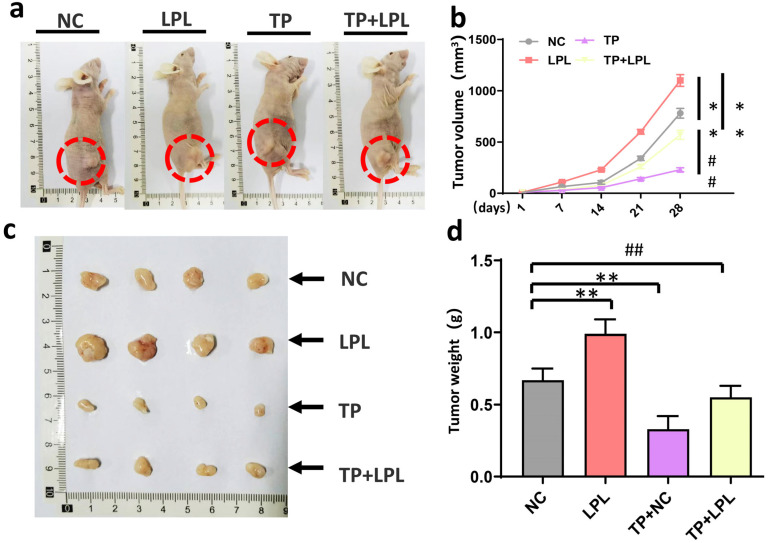
Triptolide suppressed xenograft tumor growth of HCC cells in vivo via targeting LPL. (**a**) HepG2-sh-ctrl and HepG2-sh-LPL cells were subcutaneously injected into nude mice to form xenograft tumor. Mice were treated with triptolide at the dose of 2 mg/kg/d for 5 days per week when tumors grew to ~50 mm^3^. Representative images of xenograft tumor in different groups. (**b**) Curve of tumor volume change (*n* = 8 in each group). (**c**) Representative pictures of complete xenograft tumor. (**d**) Statistical analysis of xenograft tumor (*n* = 8 in each group). *^/#^
*p* < 0.05, ** *p* < 0.01. ^##^ *p* < 0.01.

## Data Availability

The raw data supporting the conclusions of this article will be made available by the authors on request.

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
