# Peer review of "Triptolide Reduces Neoplastic Progression in Hepatocellular Carcinoma by Downregulating the Lipid Lipase Signaling Pathway"

_cancers, 2024, doi:10.3390/cancers16030550_

Round 1
Reviewer 1 Report
Comments and Suggestions for Authors
Title: Triptolide Reduces Neoplastic Progression in Hepatocellular Carcinoma by Downregulating the Lipid Lipase Signaling Pathway
This research highlights the involvement of lipoprotein lipase (LPL) in developing hepatocellular Carcinoma and the possible therapeutic role of Triptolide by downregulating LPL. Both in vivo and in vitro studies strongly support that LPL suppression decreases the HCC progression in rodent models. This manuscript is well-written and organized; however, additional concerns need to be addressed to support the findings strongly.
Comments
1. The author must explain the in silico analysis of LPL, p53 interaction with Triptolide. Does this interaction affect the activation or potentiation of p53 activity? Does the triptolide interaction with LPL decrease its enzymatic activity to reduce hepatocyte lipid loading? If so, the author must verify its inhibitory potential using an in vitro or in vivo approach.
2. From Figure 4, it is clear that the Triptolide treatment reduces the amount of LPL while increasing the p53 levels to regulate the apoptotic mechanism. At this time, it is imperative to understand the link between the LPL and p53. The author is expected to delineate the possible involvement of Triptolide on HCC signaling to reduce the expression of LPL.
Reviewer 2 Report
Comments and Suggestions for Authors
The manuscript entitled “Triptolide Reduces Neoplastic Progression in Hepatocellular Carcinoma by Downregulating the Lipid Lipase Signaling Pathway” focuses on study of mechanism of triptolide action in HCC treatment. This study scientifically sounds and may be of interest to the journal audience. However, I have some concerns which should be addressed before publication:
1. Abstract, it should be specified that HCC is the THIRD leading cause of cancer-related mortality; what is “equilibrium of lipids” – make correction; there are two contradictory sentences – the authors stated that LPL expression is upregulated in HCC and prevents HCC growth – if LPL is over-expressed it should be oncogenic.
2. Introduction, paragraph 2, the sentence “circulating lipoprotein particles” – here, types of lipoproteins and their different roles in lipid transportation should be discussed.
3. Materials and Methods, cell lines used in the work were not described; in the Results section the authors stated that they used SMMC7721 and HepG2 cell lines, but no mention regarding them in the Materials section. Why SMMC7721 cell line was used?
4. Materials and Methods, no description of molecular visualization, simulation and interaction analysis was provided. Programs and resources used for this should be provided.
5. Results section, subsection 3.1 is poorly written. The authors should avoid the phrase “according to our findings” instead use “here, we showed”. Also, the sentence in parentheses about “Mechanistic role” is unclear. Moreover, “LPL hydrolyses VLDL” is incorrect, because LPL hydrolyses triglycerides – correct this.
6. Results, subsection 3.2 – Fig. 2b, it is unclear which definite morphological and histological changes were observed in the Model, L-TP, M-TP, and H-TP? This should be described in the text and in the Figure legend.
7. Results section, subsection 3.3. In Fig. 3, it is unclear, what was a necessity to study the interaction of triptolide with p53 and LPL by simulation. No details were provided, which programs and resources were used for simulation, visualization and interaction analysis. Data given in Fig.3 is uninformative and meaningless. This part of the study constitutes a large work which required details not provided here. What does it mean: “Amino acid residues that hydrogen bond to the compound”?
8. Results section, subsection 3.3, Fig.4a –description of histological changes should be provided.
9. Discussion. In this work the authors studied changes in AST, ALT and AFP blood serum levels to assess liver functions in HCC. Therefore, the authors are recommended to provide the roles of these biomarkers in HCC diagnosis and prognosis. The following recent papers, which discuss these parameters in HCC are recommended: doi: 10.3390/biomedicines9020159, doi: 10.1186/s12876-023-02719-1.
10. English language style and grammar should be carefully checked.
Comments on the Quality of English Language
Moderate corrections are needed
Reviewer 3 Report
Comments and Suggestions for Authors
Good article, can you address few queries.
1. Triptolide (TP), has anticancer effect on many cancers including HCC. Can you please discuss the systemic toxicities limiting its clinical use.
Yu L, Wang Z, Mo Z, Zou B, Yang Y, Sun R, Ma W, Yu M, Zhang S, Yu Z. Synergetic delivery of triptolide and Ce6 with light-activatable liposomes for efficient hepatocellular carcinoma therapy. Acta Pharm Sin B. 2021 Jul;11(7):2004-2015. doi: 10.1016/j.apsb.2021.02.001. Epub 2021 Feb 4. PMID: 34386334; PMCID: PMC8343191.
2. Can you discuss the factors limiting the use of Triptolide in combination drug regimen including immunotherapy for HCC.
3. Can you discuss chemoprotective drugs for HCC.
Li Z, Yang G, Han L, Wang R, Gong C, Yuan Y. Sorafenib and triptolide loaded cancer cell-platelet hybrid membrane-camouflaged liquid crystalline lipid nanoparticles for the treatment of hepatocellular carcinoma. J Nanobiotechnology. 2021 Nov 8;19(1):360. doi: 10.1186/s12951-021-01095-w. PMID: 34749742; PMCID: PMC8576878.
Round 2
Reviewer 2 Report
Comments and Suggestions for Authors
The authors have revised the manuscript in accordance with my comments. However, some concerns remain regarding the in silico studies. This part of the study is significantly incomplete. Therefore, it is recommended to exclude these sections and to present them in another manuscript after the accomplishment of all in silico studies. The submitted manuscript does not lose anything in this case.
1. Sections 2.11 and 3.3 – Molecular docking is not enough to study the p53-TP and LPL-TP interactions. Instead, (i) Molecular dynamics simulation and (ii) Calculation of free energies of obtained complexes with analysis of interaction forces (not only hydrogen bonding, but also salt bridges and van-der-Waals interactions), and (iii) Graphical representations of the interactions are needed.
2. The authors should explain, why the corresponding crystallographic structures from PDB data base were used.
3. For all resources used including Pymol, PDB, Discovery Studio, etc. references should be provided.
4. Please check list of References – references 7, 11, 18-20, 38, 39, 43 have either incorrect format of pages or no page numbers.
Comments on the Quality of English Language
English languagh is satisfactory
